# The eco-conscious wind turbine: design beyond purely economic metrics

Helena Canet, Adrien Guilloré, and Carlo L. Bottasso

Wind Energy Institute, Technical University of Munich, 85748 Garching bei München, Germany

**Correspondence:** Carlo L. Bottasso (carlo.bottasso@tum.de)

**Abstract.**

Wind turbines are designed to minimize the economic cost of energy, a metric aimed at making wind competitive with other energy-producing technologies. However, now that wind energy is competitive, how can we increase its value for the environment and for society? And how much would environmental and societal gains cost other stakeholders, such as investors or consumers? This paper tries to answer these questions, limitedly to climate-related environmental impacts, from the perspective of wind turbine design.

Although wind turbines produce green renewable energy, they also generate various impacts on the environment, as all human endeavours. Among all impacts, the present work adopts the environmental effects produced by a turbine over its entire life cycle, expressed in terms of $CO_2$-equivalent emissions. A new approach to design is proposed, whereby Pareto fronts of solutions are computed to define optimal trade-offs between economic and environmental goals.

The new proposed methodology is demonstrated on the redesign of a baseline 3 MW wind turbine at two locations in Germany, differing for typical wind speeds but within the same energy market. Among other results, it is found that, in these conditions, a 1% increase in the cost of energy can buy about a 5% decrease in the environmental impact of the turbine. Additionally, it is also observed that in the specific case of Germany, very low specific-power designs are typically favored, because they produce more energy at low wind speeds, where both the economic and environmental values of wind are higher. Furthermore, it is found that the $CO_2$-equivalent emissions displaced by a wind turbine are one order of magnitude larger than the produced emissions.

Although limited to the sole optimization of wind-generating assets at two different locations, these results suggest the existence of new opportunities for the future development of wind energy where, by shifting the focus slightly away from a purely cost-driven short-term perspective, longer-term benefits for the environment (and, in turn, for society) may be obtained.

## 1 Introduction

The levelized cost of energy (noted here $COE_{\in}$)[1] is defined as the net-present cost of an energy-producing technology over its lifetime per MWh supplied. $COE_{\in}$ is the metric that has been traditionally used to evaluate the competitiveness of energy

---

[1] Instead of the more commonly used LCOE, the acronym $COE_{\in}$ is preferred here to emphasize the parallelism between economic and environmental metrics, as it will be clear later in the paper. Although the L is dropped from the name, in the present work economic costs are always levelized, i.e. discounted to present value. Additionally, for the lack of an instantly recognizable universal currency symbol, the European € symbol is used to indicate that this is an

sources. In recent years, the $COE_€$ from wind (and from the sun) has experienced a dramatic decrease (Roser, 2021), which in
turn has fueled an astonishing growth of wind energy and great expectations for its further expansion (Veers et al., 2019). About
a decade ago, the International Energy Agency (IEA) Wind Task 26, which focuses on the cost of wind energy, identified a key
driver for the future development of wind technology: the ability of generating cost parity – without direct policy support –
with conventional sources, in a broad range of conditions and locations (Lantz et al., 2012). This indeed has largely happened
and the evolution of wind energy technology continues at a fast pace, to the point that even offshore wind is rapidly marching
towards subsidy-free competitiveness (Jansen et al., 2020). The decrease in $COE_€$ from wind has been partially driven by
technological advancements, which have led to more reliable turbines characterized by higher hub heights and larger rotor
diameters and, most importantly, much improved capacity factors. Additionally, economies of scale, increased competitiveness
and an improved maturity of the sector have also contributed to the fall of $COE_€$ witnessed in recent years (IRENA, 2021).

$COE_€$, however, paints only a partial view of a situation that is much more complex and articulated than what appears
through cost alone (Joskow, 2011). A more holistic picture of the overall effects of renewable energies in general, and of wind
in particular, can only be obtained when looking beyond cost metrics. Indeed, the urgency created by climate change, energy
security and independence could not be clearer, as stressed by the headline news coming from all over the world every day.

In fact, the future participation of wind power in the energy market and, more broadly, its societal role will not only be
shaped by its relative competitiveness, but also by its *value* (Beiter et al., 2021). The word *value* is generally understood in the
literature as a synonym for *economic value*, which is a measure of the benefit provided to an actor by some good or service.
In reality, in the case of an energy-generating technology, the concept of value is extremely broad. Leaving aside aspects such
as energy security and independence, which are of crucial importance but also beyond the scope of the present analysis, it is
worth noticing that the value of an energy technology cannot be quantified per se, because it depends on the interactions of that
technology with the system in which it operates (Mai et al., 2021). For instance, the total system-value of an asset can be seen
as the sum of different system-value components – including energy value, capacity value, ancillary service value and others
(Mai et al., 2021). Additionally, due to supply and demand variability, the market price of electricity can vary widely, with high
wholesale prices during peak demand times, which however can reach down to even negative values when large amounts of
renewable energy are available in the grid. This fact, in addition to transmission and storage constraints, makes the economic
value of electricity time- and location-specific (Hirth, 2012).

The importance of value has not gone unnoticed to the recent literature, and a range of options for increasing the economic
value of wind energy have been explored. For instance, the geographic location of wind plants – and, more in general, of
variable renewable energy plants – and the diversification of the energy mix are two strategies that can be used to this effect
(Hirth and Mueller, 2016). Additionally, even the design characteristics of wind-generating assets (which is the focus of the
present work) can change when considering value, rather than simply cost. In fact, some wind turbine design parameters –
in particular hub height and specific power (i.e. rated power divided by rotor swept area) – can have a significant effect on
economic value, as shown by Hirth and Mueller (2016); Lantz et al. (2017); Swisher et al. (2022), among others.

---

economic cost term, to distinguish it from a parallel environmental cost defined later. However, it is clear that economic costs could be expressed also in other
currencies besides the euro.

Economic cost and value, however, are actor-centric metrics, which mostly capture the investor point of view and, in turn, also the price eventually paid by the end consumer. Additionally, cost and value are short-term metrics: cost evolves rapidly from year to year, whereas value changes on even much faster time scales of minutes/hours. In this sense, economic cost and value – if used alone – seem to be rather myopic metrics for the design of a wind turbine. Indeed, time is ripe for looking beyond the benefit of the single actor and beyond short-term effects: wind energy should evolve to also take into account its broad and long-term impacts on the environment and at the societal level. It is a major ambition of this paper to bring this new point of view to the design of wind turbines. Clearly, the same philosophy can also be applied to the design of wind plants and, more in general, to the design of the whole energy system.

From this broader perspective, the overarching goal of design becomes the alignment of short-term economic needs with long-term sustainable development goals. In fact, while it is necessary to enhance the economic value of wind energy to increase its competitiveness today, it is also our moral duty to improve the value of this technology for the environment and for society now and into the future.

How can these broader goals be achieved? What are the new metrics that should be used to capture these longer-term effects? How can value be defined beyond its current economic meaning? If new turbines were designed according to these principles, how different would they be from standard $COE_\euro$-driven designs? And how much would a environmental/societal-level gain cost in terms of $COE_\euro$? These are some of the questions that are in need of answers, and that the present paper is trying to address, albeit in a preliminary and certainly yet incomplete form.

There are undoubtedly several different options for including long-term societal and environmental effects in the design of wind turbines. For example, societal metrics could capture – among others – health, safety, security, acceptance, cultural preservation, employment, education, etc. Metrics relevant to the environment could include impacts on climate change, the depletion of resources (such as water, land, rare and scarce materials), energy use, effects on wildlife and biodiversity, and others. This study focuses exclusively on the impact exerted by wind technology on the environment in terms of greenhouse gas (GHG) emissions. While GHG emissions clearly do not capture all effects of wind energy, they do provide for a major and quantifiable impact with long-term consequences. As long as they are quantifiable through some appropriate metric, other societal and environmental impacts could be included in a future even more general approach than the one presented here.

At first glance, it might seem unusual to speak about GHG emissions in the context of wind energy. After all, a wind turbine is an eco-friendly machine by definition, which captures kinetic energy from wind to produce electricity without directly releasing pollutants into the environment. Additionally, the deployment of each new wind turbine displaces a certain amount of GHG emissions, because the output of other more polluting energy sources can be correspondingly reduced. However, even wind turbines do have an environmental cost – as indeed all human activities –, and non-negligible amounts of GHGs are emitted throughout the different stages of their life. For example, the production of the large amount of steel required for the tower, or the extraction of raw materials – such as the rare-earth elements present in the generator –, do have significant environmental impacts; additionally, the end-of-life (EOL) treatments of components with limited recyclability, such as blades largely made of reinforced thermoset polymers, do release polluting emissions into the atmosphere. More in general, all stages of the life-cycle of a turbine, from the extraction of raw materials all the way to the eventual disposal/recycling/repurposing of its components,

generate impacts that can be quantified in terms of $CO_2$-equivalent emissions. Given its importance, it is no surprise that the evaluation of the environmental cost of wind turbines is the subject of various recent studies, including Al-Behadili and El-Osta (2015) and Ozoemena et al. (2018), among others. In addition to representing a meaningful metric per se, GHG emissions

can also be turned into economic costs by using the societal cost of carbon (SCC), which is an estimate of the net present value of monetized social damages occurring from the emission of an additional metric ton of $CO_2$ (National Academies of Sciences, Engineering and Medicine, 2017; Gillingham and Stock, 2018). However, care should be exercised when using SCC, as it can take a broad range of values, depending on the underlying assumptions and models (IPCC, 2007; Ricke et al., 2018; Kikstra et al., 2021).

While several publications propose metrics that capture the economic profitability of a wind turbine (Ueckerdt et al., 2013; Simpson et al., 2020; Mai et al., 2021), no metrics are yet available to describe the environmental cost and value of wind energy. To address this gap, this work introduces novel eco-conscious metrics that mirror existing economic ones. These metrics are then used within a multi-objective design framework, which sizes some macroscopic parameters of a wind turbine (here rotor diameter and hub height) to find optimal trade-offs between economic and environmental perspectives.

The eco-conscious metrics are defined based on a life-cycle assessment (LCA) method, which has the added benefit of breaking down the contribution to the overall GHG emissions of a wind turbine by its components, materials and life-cycle stages. This way, a ranking of the most harmful aspects of a design is readily obtained, revealing new opportunities and highlighting the most promising pathways for further mitigating GHG emissions beyond what is possible by sizing alone (Guilloré et al., 2022).

The paper is organized as follows. Section 2 defines metrics that quantitatively measure the cost and value of a wind-generating asset, both from the economic and the environmental perspectives. Next, Sect. 3 describes the methods that were used here to estimate the design metrics. In addition to standard energy production, mass, and cost models, this section describes and validates an LCA model that estimates the $CO_2$-equivalent emissions produced during each stage of the life and by each component of a wind turbine. The design approach is formulated in this same section, in terms of single- and multi-objective

constrained optimization problems. The new proposed methodology is exercised in Sect. 4, by redesigning a baseline 3 MW wind turbine at two different locations in Germany, one in the north and the other in the south of the country, characterized by different wind resources but within the same electricity market. The results are analyzed by looking at the trade-offs between economic and environmental metrics, and at the change in the design characteristics of the optimal turbines with respect to a standard $COE_{\in}$-driven baseline. Finally, Sect. 5 summarizes the main findings of this study and offers an outlook towards

future work.

## 2    Design metrics from economic and environmental perspectives

This section describes metrics for the preliminary design of an energy-generating unit using three common concepts: cost, value, and net value. In the economic context, *cost* indicates the monetary expense incurred for making a product or service, whereas *value* (or *revenue*) is a measure of the monetary benefit brought by that good or service. The difference between cost

and value is termed *net value* (or *profit*). The good or service considered here is the production of energy. The three terms cost, value and net value will be used with two different connotations: the classical *economic* one, when relating to money, and the *environmental* one, when relating to the GHGs emitted in the lifetime of an asset. Besides this climate-change-related impact category, it is clear that other environmental impacts could be considered through appropriate quantitative metrics. Table 1 summarizes the metrics defined in the next pages, categorized in terms of the three concepts of cost, value and net value, and from the two economic and environmental perspectives. These metrics are applicable to both single generating units (e.g., a wind turbine) or a plant (e.g., a wind farm), although the present work focuses only on the former case. For metrics not based on cost, value or net value, the reader is referred to Ueckerdt et al. (2013); Simpson et al. (2020) and Mai et al. (2021).

**Table 1.** Overview and parallelism of economic and climate-change-related environmental metrics.

|  | Economic perspective $[\text{\euro}/\text{MWh}]$ | Environmental perspective $[\text{kgCO}_2\text{eq}/\text{MWh}]$ |
|---|---|---|
| **Cost** | $\text{COE}_{\text{\euro}}$ | $\text{COE}_{\text{CO2}}$ |
| **Value** | $\text{VOE}_{\text{\euro}}$ | $\text{VOE}_{\text{CO2}}$ |
| **Net value** | $\text{NVOE}_{\text{\euro}}$ | $\text{NVOE}_{\text{CO2}}$ |

## 2.1 Economic perspective

The economic perspective results from an actor-centric point of view, e.g. the investor or the consumer, where the focus is primarily driven by short-term economic forces, such as profit or out-of-pocket expenditure.

### 2.1.1 Economic levelized Cost of Energy ($\text{COE}_{\text{\euro}}$)

$\text{COE}_{\text{\euro}}$ is an estimate of the average net-present cost of each unit of energy produced over the lifetime of a generating asset. As such, this metric is widely used to assess the competitiveness of different energy technologies. $\text{COE}_{\text{\euro}}$ is formally defined as the ratio of the discounted lifetime costs and the discounted generated energy (Aldersey-Williams and Rubert, 2019; Duffy et al., 2020), i.e.

$$\text{COE}_{\text{\euro}}\left[\frac{\text{\euro}}{\text{MWh}}\right] = \frac{\text{Economic costs}}{\text{Energy production}} = \frac{\sum_{y=1}^{Y}\frac{C_y^{\text{CAPEX}}+C_y^{\text{OPEX}}}{(1+d)^y}}{\sum_{y=1}^{Y}\frac{E_y}{(1+d)^y}}, \tag{1}$$

where the subscript $(\cdot)_y$ indicates the $y$-th year and $Y$ is the lifetime in years, while $C^{\text{CAPEX}}$ are the capital costs, $C^{\text{OPEX}}$ are the operating and maintenance costs, $E$ is the asset-generated energy, and finally $d$ is the discount rate.

Capital costs include all expenditures incurred to manufacture the asset, while the operating and maintenance costs include all expenditures necessary for running the asset and maintaining it in working conditions (Joskow, 2011; Mai et al., 2021). The discount rate is the interest rate used to determine the present value of future cash flows and, therefore, expresses the time value of money. The discount rate is often affected by significant uncertainties, which in turn may impact $\text{COE}_{\text{\euro}}$.

### 2.1.2 Economic levelized Value of Energy (VOE$_\text{€}$)

VOE$_\text{€}$ is an estimate of the average net-present economic value of each unit of energy produced over the lifetime of a generating asset (Mai et al., 2021). Similarly to COE$_\text{€}$, VOE$_\text{€}$ is defined as

$$\text{VOE}_\text{€} \left[ \frac{\text{€}}{\text{MWh}} \right] = \frac{\text{Economic value}}{\text{Energy production}} = \frac{\sum_{y=1}^{Y} \frac{V_y}{(1+d)^y}}{\sum_{y=1}^{Y} \frac{E_y}{(1+d)^y}}. \tag{2}$$

The total revenue $V_y$ generated by the asset in the $y$-th year is computed as a function of time $t$ as

$$V_y = \int_{t=0}^{T_y} p(t)P(t)\,\mathrm{d}t, \tag{3}$$

where $T_y$ is the year duration, $p(t)$ is the spot market price in €/MWh, and $P(t)$ is the power produced by the unit at time instant $t$. Alternatively, the same quantity can be estimated as a function of wind speed $U$ as

$$V_y = T_y \int_{U_i}^{U_o} p_y(U)P(U)W_y(U)\,\mathrm{d}U, \tag{4}$$

where $U_i$ and $U_o$ are respectively the cut-in and cut-out wind speeds, $P(U)$ is the turbine power curve, while $p_y(U)$ and $W_y(U)$ are respectively the spot market price of energy and the Weibull probability density function at the site where the asset is installed in the year $y$.

### 2.1.3 Economic Net Value of Energy (NVOE$_\text{€}$)

NVOE$_\text{€}$ is defined as the difference between VOE$_\text{€}$ and COE$_\text{€}$ (Mai et al., 2021), i.e.

$$\text{NVOE}_\text{€} \left[ \frac{\text{€}}{\text{MWh}} \right] = \frac{\text{Economic value} - \text{Economic cost}}{\text{Energy production}} = \text{VOE}_\text{€} - \text{COE}_\text{€}. \tag{5}$$

## 2.2 Environmental perspective

Adopting an environmental perspective, the goal is no longer to achieve the cheapest energy in the short term, but rather the most sustainable one in the long term. The metrics presented here mirror the ones defined in the previous section. However, instead of considering the economic perspective, these novel metrics focus on the climate-change-related environmental impact, which is quantified in terms of $CO_2$-equivalent emissions.

As money is attributed a time value through the discount rate, even impacts could in principle be discounted, because emissions produced/displaced today might have a different effect from the ones of tomorrow. Indeed, time horizons are included in the estimation of the Global Warming Potential (GWP) that is used to convert the effects of different gases into equivalent $CO_2$ climate impacts (IPCC, 2007). However, discount rates for $CO_2$-equivalent emissions are at present not available, and would probably be subjected to high uncertainties; therefore, discount rates were not considered in the definition of these environmental-based metrics.

### 2.2.1 Environmental Cost of Energy (COE$_{CO2}$)

COE$_{CO2}$ represents an estimate of the average environmental cost of each unit of energy produced over the lifetime of a generating asset:

$$\text{COE}_{\text{CO2}}\left[\frac{\text{kg CO}_2\text{eq}}{\text{MWh}}\right] = \frac{\text{Environmental cost}}{\text{Energy production}} = \frac{\sum_{m=1}^{M} Q_m}{\sum_{y=1}^{Y} E_y}, \tag{6}$$

where Q$_m$ is the CO$_2$-equivalent GHG emissions during life-cycle stage $m$, and $M$ is the total number of life-cycle stages of the asset, from the extraction of the raw materials all the way to EOL treatments. COE$_{CO2}$ is the environmental counterpart of COE$_{€}$, with the difference that decommissioning and EOL costs are generally not considered in the definition of the latter. Similar definitions of COE$_{CO2}$ have been given elsewhere using different names, as for example *CO$_2$ Intensity* (Tremeac and Meunier, 2009), *Emission Factor* (Koffi et al., 2017), *Carbon Footprint* (Hauschild et al., 2018), and *Global Warming Potential* (Ozoemena et al., 2018). The present name and acronym is preferred here, because it helps convey the parallelism between the economic and environmental perspectives.

### 2.2.2 Environmental Value of Energy (VOE$_{CO2}$)

VOE$_{CO2}$ is the counterpart of VOE$_{€}$, and it is defined as the average environmental value per unit of energy generated by an asset over its lifetime:

$$\text{VOE}_{\text{CO2}}\left[\frac{\text{kg CO}_2\text{eq}}{\text{MWh}}\right] = \frac{\text{Environmental value}}{\text{Energy production}} = \frac{\sum_{y=1}^{Y} V_y^{\text{env}}}{\sum_{y=1}^{Y} E_y}. \tag{7}$$

The environmental value is quantified here in terms of the CO$_2$-equivalent emissions that are displaced in the grid by the energy-producing asset. At time $t$, the energy mix is characterized by $G_t$ generating technologies, each producing a certain power $P_g(t)$. The activation of a renewable generating unit that produces a power output $P(t)$ displaces some output $P_g^{\text{dis}}(t)$ of the $g$-th generating technology, such that $P(t) = \sum_{g=1}^{G_t} P_g^{\text{dis}}(t)$. Despite the activation of a renewable generating unit, the time-dependent total power in the grid remains the same, as it is driven by demand. As a consequence, an environmental value $V_y^{\text{env}}$ is generated over the time duration $T_y$, which is equal to the amount of displaced emissions, i.e.

$$V_y^{\text{env}} = \int_{t=0}^{T_y} \sum_{g=1}^{G_t} f_g(t) P_g^{\text{dis}}(t)\, dt. \tag{8}$$

The emission factor $f_g$ quantifies the environmental impact of each generating technology in the mix. This quantity depends on time, because it is related to the operational conditions of the generating technology. For instance, operating a fossil-fueled plant at partial load has an efficiency penalty that increases the fuel consumption and the GHG emissions per unit of generated energy (Silver-Evans et al., 2012; Thomson et al., 2017). For simplicity, here each given technology $g$ is associated with an average time-independent emission factor defined as

$$f_g = \frac{Q_g}{E_g}, \tag{9}$$

where $Q_g$ indicates the average $CO_2$-equivalent GHG emissions caused by the production of an amount of energy $E_g$.

The actual displacement of grid emissions is a complex time-dependent phenomenon (Hawkes, 2010; Thomson et al., 2017; Boeing et al., 2019). In fact, the only emissions that will be displaced are the ones of generators operating on the margin, i.e. the last generators needed to meet demand at a given time that are capable of rapidly adapting their power generation in response to a change in demand (Silver-Evans et al., 2012; Seckinger, 2021). Therefore, the actual displacement of grid emissions is determined by these marginal generators, which in turn depend on time-variable factors such as power demand, resource availability (e.g., wind speed and solar irradiation), or availability of other generation technologies. For simplicity, here it is assumed that all generating technologies are displaced equally, i.e. $P_g^{\mathrm{dis}}(t)/P_g(t) = P(t)/\sum_{g=1}^{G_t} P_g(t)$, for each generating technology $g$ at each time $t$. This is a conservative approach that is generally used to estimate emission displacements, and which has been shown to underestimate the real displacement potential of wind energy (Hawkes, 2010; Silver-Evans et al., 2012; Thomson et al., 2017). Under this hypothesis, the environmental value writes

$$V_y^{\mathrm{env}} = \int_{t=0}^{T_y} \frac{\sum_{g=1}^{G_t} f_g P_g(t)}{\sum_{g=1}^{G_t} P_g(t)} P(t)\,\mathrm{d}t = \int_{t=0}^{T_y} f_{\mathrm{grid}}(t) P(t)\,\mathrm{d}t. \tag{10}$$

The expression on the right hand side of the equation considers the whole grid as one aggregated generating unit, characterized by one equivalent time-dependent system-average emission factor $f_{\mathrm{grid}}(t)$, which reflects the composition of the energy mix at each time instant (Thomson et al., 2017; Seckinger, 2021).

As for economic value $V_y$, also environmental value $V_y^{\mathrm{env}}$ can be estimated as a function of wind speed, instead of time, by the following expression

$$V_y^{\mathrm{env}} = T_y \int_{U_i}^{U_o} f_{\mathrm{grid}}(U) P(U) W_y(U)\,\mathrm{d}U. \tag{11}$$

### 2.2.3 Environmental Net Value of Energy (NVOE$_{CO2}$)

NVOE$_{CO2}$ is the counterpart of the economic metric NVOE$_€$, and it is defined as the difference between the environmental value of energy and cost of energy, i.e.

$$\mathrm{NVOE_{CO2}} \left[ \frac{\mathrm{kg\ CO_2 eq}}{\mathrm{MWh}} \right] = \frac{\mathrm{Environmental\ value - Environmental\ cost}}{\mathrm{Energy\ production}} = \mathrm{VOE_{CO2}} - \mathrm{COE_{CO2}}. \tag{12}$$

### 2.2.4 Future economic Societal Savings (FSS)

FSS estimates the future societal savings enabled by the displacement of GHG emissions, and writes

$$\mathrm{FSS} \left[ \frac{€}{\mathrm{MWh}} \right] = \mathrm{SCC} \cdot \mathrm{NVOE_{CO2}}. \tag{13}$$

The societal cost of carbon (SCC) is the present discounted monetary value of the future damage caused to the environment by one metric ton increase in $CO_2$-equivalent emissions (National Academies of Sciences, Engineering and Medicine, 2017). The quantification of SCC is clearly not a straightforward task. Indeed, the literature reports a large range of values (Ricke et al.,

2018; Kikstra et al., 2021), mostly due to different assumptions on climate sensitivity, economic and non-economic impacts, and response lags, among others (IPCC, 2007). Additionally, $NVOE_{CO2}$ depends on $VOE_{CO2}$ that, as previously argued, is based on the simplifying assumption that all generation technologies are equally displaced by wind power; since this is hardly exactly true in practice, further uncertainties are introduced in the estimation of FSS. Notwithstanding these limitations, FSS is considered here because it allows, to some extent, to turn climate-change-related environmental effects into societal ones.

However, societal impacts are much broader than this, and additional ad hoc metrics should be used to capture their full spectrum.

## 3    Methods

This section describes the eco-conscious design of wind turbines, formulated as a constrained multi-objective optimization problem based on a number of interconnected underlying models. Figure 1 shows a schematic representation of the workflow

and of its main components.

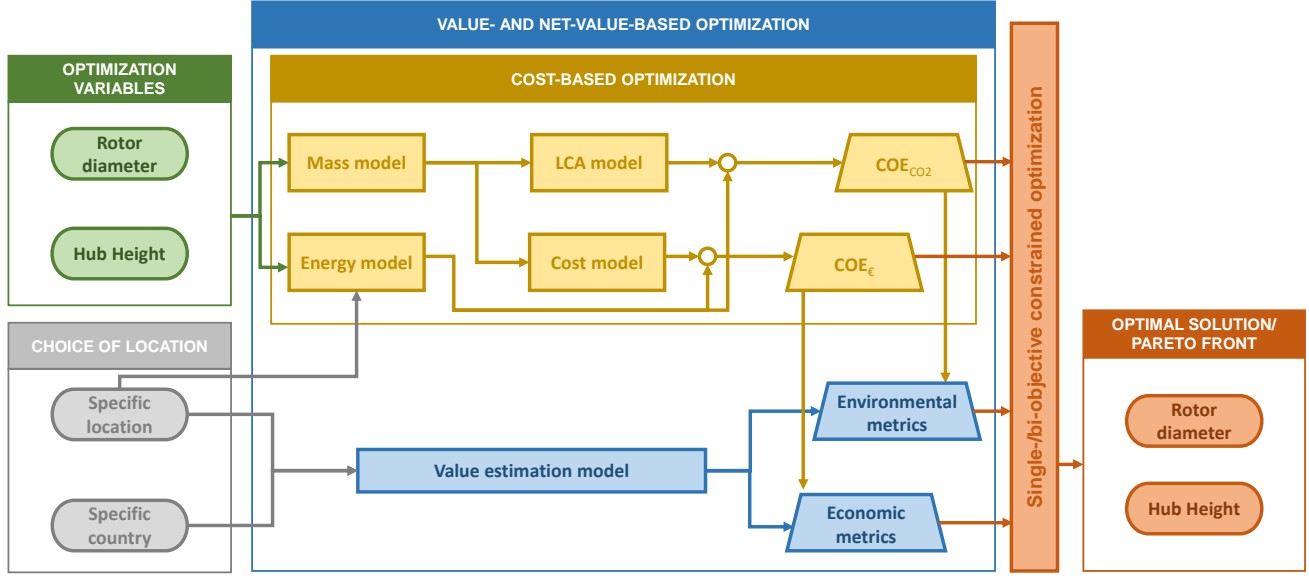

**Figure 1.** Schematic illustration of the workflow for the eco-conscious multi-objective design optimization of wind turbines. Rounded squares represent variables, squares are models and trapezoids are merit functions.

### 3.1 Energy model

The energy $E_y$ produced by a wind turbine at a specific location in the year $y$ is computed as

$$E_y = f_a f_p f_w T_y \int_{U_i}^{U_o} P(U) W_y(U) \, \mathrm{d}U. \tag{14}$$

Three correction coefficients are included in the formula. The availability factor $f_a$ accounts for failures, maintenance, and curtailment time, and it is set to the typical value of 0.98 (Vestas, 2011, 2013a, b; Pfaffel et al., 2017). The performance factor $f_p$ considers different sources of losses due to turbulence, gusts, wakes, blade soiling/erosion, etc., and it is set to the value of 0.65, which is the lower limit of the range indicated in Lantz et al. (2017). The wind factor $f_w$ accounts for possible deviations of the wind resource from the Weibull distribution, for example due to yearly variability (Lantz et al., 2017). Since the present analysis is based on actual historical data, which already includes any variability of the resource, $f_w$ is set to the value of one.

### 3.2 Mass model

The mass estimation model is composed of three sub-models. The mass of the various turbine components is based on the 2017 NREL mass model (NREL, 2021), which is an updated version of the 2006 cost and scaling model (Fingersh et al., 2006). The inputs to the mass model are rated power, hub height and rotor diameter. The gearbox mass is estimated based on the maximum generator torque, which was computed assuming a fixed maximum tip speed of 80 ms$^{-1}$. Based on the mass of the single components, a material breakdown model defines a bill of materials. This model is based on information sourced from several references (Rydh et al., 2004; Vestas, 2011, 2013a, b; Demir and Taskin, 2013; Haapala and Prempreeda, 2014; Ozoemena et al., 2018), and includes 15 different material types: glass fiber, carbon fiber, epoxy resin, sandwich foam, alloyed steel, unalloyed steel, galvanized steel, copper, aluminium, PVC and other plastics, rubber, paint and coating, Neodymium permanent magnet (NdFeB), electronics, and concrete. Finally, a waste factor model estimates the quantity of material that is wasted during the different stages of the component lifetime. Waste factors for fiberglass, epoxy resin, foam, rubber, paint and coating are modeled according to Bortolotti et al. (2019), while a factor of 5% is considered for the other materials.

The use of mass and scaling models is one of the various approximations of the present approach. More precise estimates of masses and bills of materials would clearly be possible by using detailed sizing procedures (Bortolotti et al., 2016; NREL, 2021). This level of complication and computational cost was however not deemed necessary for capturing general trends, which is the main goal here.

### 3.3 LCA model

LCA is a normed scientific methodology to exhaustively assess the environmental impacts of a product or a service, over its entire lifetime from cradle to grave. Here LCA is performed by an in-house-developed literature-sourced model that follows the environmental management standards of the International Organization for Standardization (ISO), according to ISO 14040 and ISO 14044 (Wolf et al., 2012; Hauschild et al., 2018).

The objective of the LCA model is to assess the complete life-cycle GHG emissions associated with the production of one functional unit, which in this case is 1 kWh of electricity. Emissions are broken down in terms of life-cycle stages, components and materials. Only climate-change-related environmental impacts are considered, and other effects such as human toxicity, eco-toxicity, acidification or resource depletion are excluded.

The model is formulated in a parametric way, i.e it is not specific to a given wind turbine type, and it is generally applicable to contemporary onshore variable-speed horizontal-axis technology. It is assumed that the turbine is installed in Europe between the years 2015 and 2025, and has a lifetime of 20 years. The machine is composed by rotor, nacelle, drivetrain, tower and foundations, and the elements within these components (e.g., the generator). Connection to the grid, storage or other equipment and devices are outside of the scope of this model.

The processes involved in each one of the life-cycle stages are modeled based on typical scenarios from Rydh et al. (2004); Vestas (2011, 2013a, b); Demir and Taskin (2013); Haapala and Prempreeda (2014); Ozoemena et al. (2018), among others. Emission factors are based on Ecoinvent IPCC 2013 (Myhre et al., 2013; Ecoinvent, 2019; Bourgault, 2019).

This LCA method considers the atmospheric emissions of all gases that are recognized to have a greenhouse effect, including $CO_2$, $CH_4$, $N_2O$ and fluorinated gases. For each one of these gases, the mass of $CO_2$ that would have the same greenhouse effect is defined and used as a measure of impact (Myhre et al., 2013; Bourgault, 2019).

### 3.3.1 Life-cycle stages

This section briefly defines the life-cycle stages considered in the present work, and the assumptions taken in each of them. For further details, the reader is referred to Guilloré et al. (2022).

- Life-cycle stage 1: Raw material extraction and processing. This stage accounts for the environmental impact upstream of the purchasing of a unit of ready-to-use material for manufacturing. Raw material extraction and processing emissions are modelled according to Ecoinvent (2019), assuming that all materials derive from primary sources – i.e. there is no recycled content.

- Life-cycle stage 2: Transportation of raw materials to manufacturing sites. This stage considers both direct emissions caused by the burning of transportation fuel, and indirect emissions produced in the life-cycle of the fuel from well to tank. Indirect emissions from the production of the transportation technology itself are also included. Based on Vestas (2011, 2013a, b), it is assumed that all materials are transported over a distance of 600 km to the manufacturing site, except for concrete, which is only transported over a distance of 50 km. Emission factors for transportation are considered from Ecoinvent (2019), assuming that materials are transported by freights and lorries heavier than 32 t, with EURO4 exhaust emissions (Spielmann et al., 2007).

- Life-cycle stage 3: Wind turbine component manufacturing. This stage considers the environmental impact of the energy consumed for the transformation of the materials into wind turbine components. The model includes the upstream environmental impact of the consumed energy – which is generally electricity from the grid, whose impact in turn de-

pends on the specific electricity mix. Manufacturing emissions are obtained from several sources (Song et al., 2009; Hill and Norton, 2018; Ecoinvent, 2019).

- Life-cycle stage 4: Transportation of the components to the wind plant site. For this stage, the same assumptions on transportation vehicles of life-cycle stage 2 are taken, adding ship transport. Assumptions on transportation distances are modeled according to Vestas (2011, 2013a, b).

- Life-cycle stage 5: Assembly and installation of the wind turbine. This life-cycle stage considers the direct and indirect emissions from the assembly and installation of the different wind turbine components. It is assumed that a hydraulic crane is required for 16 hours of work (Rydh et al., 2004; Ozoemena et al., 2018).

- Life-cycle stage 6: Operation and maintenance (O&M). This stage considers different impacts related to operation and maintenance, and is defined according to Rydh et al. (2004); Vestas (2011, 2013a, b); Demir and Taskin (2013); Haapala and Prempreeda (2014); Ozoemena et al. (2018). The GHG emitted during O&M are determined as the sum of the emissions related to the turbine lubricant oil change, to the use of an inspection van and maintenance crane, and related to the replacement of components, as detailed next:

    - Lubricant Oil. The oil employed for the regular change of gearbox lubricant is considered. Assumptions are taken according to Rydh et al. (2004), Haapala and Prempreeda (2014) and Ozoemena et al. (2018).

    - Inspection van. It is assumed that a roundtrip from the maintenance base is required every 6 months (Ozoemena et al., 2018) with a diesel passenger car of emission category EURO4 (Spielmann et al., 2007).

    - Maintenance Crane. It is considered that heavy crane machinery is required for a total of 8 hours over the turbine lifetime (Ozoemena et al., 2018).

    - Replacements of components. All components may be subjected to failures, and generally several parts need to be replaced over the lifetime of a wind turbine. Failure rates are modeled according to Tremeac and Meunier (2009); Demir and Taskin (2013); Haapala and Prempreeda (2014); Ozoemena et al. (2018). Life-cycle stages 1 to 5 are used to estimate the emissions resulting from the spare components that need to be replaced. Additionally, the impact of the transport of the replacement components to the site is doubled, to account for the trip back with the replaced components.

- Life-cycle stage 7: Decommission and transportation of parts. This life-cycle stage considers 16 hours of crane work, as described in Rydh et al. (2004) and Ozoemena et al. (2018). The same assumptions taken for life-cycle stage 4 are used also here to estimate the emissions caused by the transportation of the parts to their EOL treatment centers.

- Life-cycle stage 8: EOL treatment. The EOL scenario is a key stage in the life-cycle of a wind turbine. Three treatments are considered: recycling, incineration, and landfilling. In accordance with ISO 14044 (Wolf et al., 2012; Hauschild et al., 2018), this work adopts the closed-loop material cycle approach, where full credit is given to the emissions of life-cycle stage 1 for materials that are recycled at the end of the component life. This way, recycled materials are considered to have

a negative impact, and thus represent environmental benefits. In reality, this full closed-loop recycling credit scenario represents only a limit case, because recyclability pathways are complex and need to consider material degradation and other effects. When these are considered, emission burdens can be more precisely allocated between first and second-life users, for example through the *Circular Footprint Formula* recently proposed by Zampori et al. (2019); such refinements of the approach are left to forthcoming work. Metals – steel, copper, and aluminium – have high recyclability rates, as shown in Fig. 2 (Tremeac and Meunier, 2009; Vestas, 2011, 2013a, b; Haapala and Prempreeda, 2014; Schmid, 2020). On the other hand, there is no mature technology yet for the recycling of thermoset glass-fiber reinforced polymers (GFRP), which are currently incinerated or landfilled (Schmid, 2020; Beauson et al., 2022). The overall EOL impact is the sum of the recycling, incineration and landfilling environmental impacts. This quantity can either be positive or negative, depending on whether or not the recycling benefits outweigh the incineration and landfilling environmental impacts. Whether blade EOL is by incineration or landfilling depends on the legislation of the country, which is far from uniform even across Europe; for example, some countries like Germany or the Netherlands forbids the landfilling of composites altogether (WindEurope, 2020; Beauson et al., 2022). On the contrary, the landfilling of wind turbine blades is widespread in the US, due to its low cost and the lack of specific legislation that prohibits it (Beauson et al., 2022; Ramirez-Tejeda et al., 2017). As the present study is located in Germany, a scenario of $100\%$ incineration is used here. Clearly, this is a simplification of a much more complex EOL reality, because other solutions – such as cement co-processing – are used in Germany for the EOL of blades (WindEurope, 2020; Beauson et al., 2022).

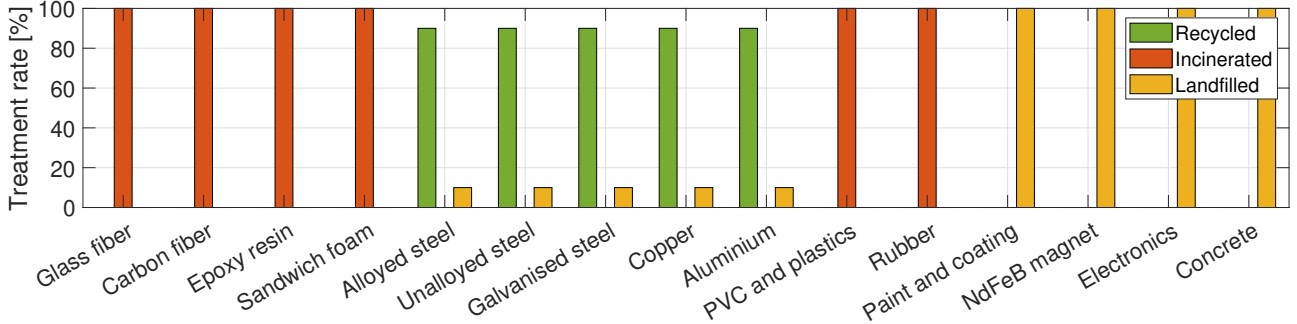

**Figure 2.** EOL treatment rates (by mass) for various materials types.

### 3.3.2 Validation of the LCA model

The LCA model was validated against results published by Schleisner (2000); Tremeac and Meunier (2009); Vestas (2011, 2013a, b); Al-Behadili and El-Osta (2015); Ozoemena et al. (2018), as shown in Fig. 3.

In general, there is a good match between previous studies and the present model. Differences arise due to non identical hypotheses and assumptions in life-cycle scenarios, bill of materials, energy production, or other aspects of the models. For example, Vestas (2011, 2013a, b) consider an average EOL treatment for composites of 50% incineration and 50% landfilling,

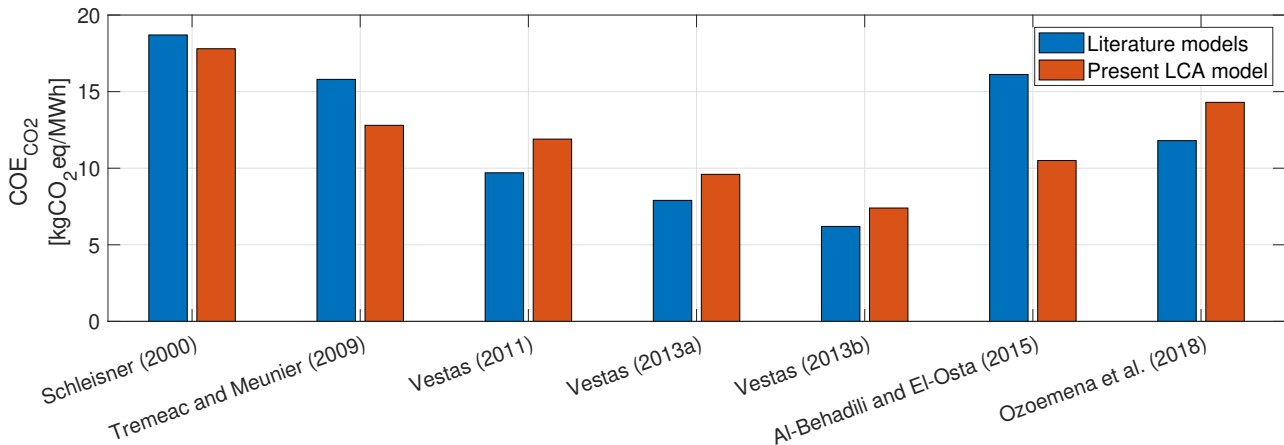

**Figure 3.** Comparison of the environmental impact obtained with the present LCA model and with results sourced from the literature.

while a scenario of 100% incineration is assumed in the present study; the study of Al-Behadili and El-Osta (2015) is located in Lybia, while the present one in Europe; additionally, Ozoemena et al. (2018) apply a recycled content approach and therefore do not consider any recycling credit at end of life. Additionally, several publications do not thoroughly detail the assumptions they are based on, or the processes considered in the different life-stage cycles, which hinders an exact comparison.

### 3.4 Cost model

Costs are based on the 2015 NREL cost model (NREL, 2021), converted to 2017€ values. The model estimates the initial capital costs and O&M costs. Initial capital costs include rotor, nacelle, drivetrain, tower and foundations, as well as balance of station (BOS) costs, including transportation, assembly and installation. Additional BOS-related costs such as engineering, permitting, and grid connection are excluded, as their environmental impact is not considered in the present LCA model. Annual operating expenses include O&M costs, whereas land lease costs are not considered.

### 3.5 Value estimation model

This model estimates the economic and environmental value of a wind turbine, for a specific location and a specific time frame, as illustrated in Fig. 4.

The estimation of economic value is based on historical data, using Eq. (4). Time series of spot market price were correlated with time series of wind speed at a specific location and hub height, resulting in the price-wind model $p_y(U)$. Similarly, the environmental value was estimated with Eq. (11), where the grid average emission factor $f_{grid}(U)$ was computed based on the energy mix time history of the country, or region, where the turbine is located. The average emission factor of each generation technology in the mix was obtained from Ecoinvent (2019), and only considers operational emissions (Thomson et al., 2017;

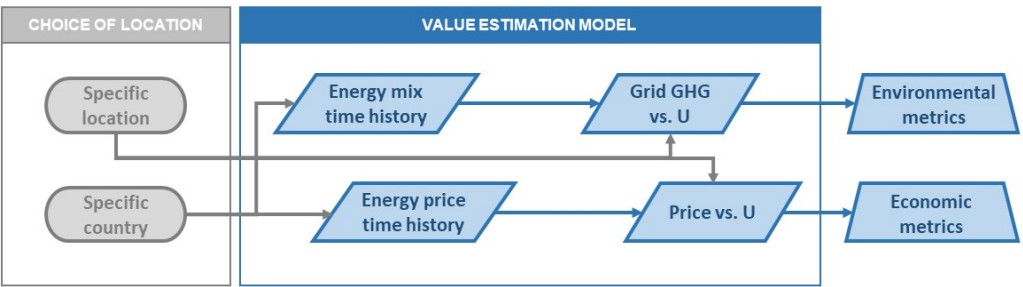

**Figure 4.** Workflow of the value estimation model of Fig. 1. Rounded squares represent variables, squares are models, rhombuses are data, and trapezoids are metrics.

Boeing et al., 2019). Wind speed time histories were adjusted to the turbine hub height based on the site mean shear, and were used to estimate the Weibull distribution, and to adjust the price-wind $p_y(U)$ and grid-average emission $f_{grid}(U)$ factors.

### 3.6 Optimal design problems

In this work two different design problems are considered, based on either a single- or a two- objective constrained optimization.
In both cases, the problem is formulated as:

$$\min_{\mathbf{p}} \quad J(\mathbf{p}), \tag{15a}$$

such that:   $\mathbf{c}(\mathbf{p}) \leq 0,$ (15b)

where $J$ is the cost function, chosen among the design metrics of Sect. 2, $\mathbf{p} = (D, H)$ are the design variables, with $D$ the rotor diameter and $H$ the hub height. Finally, $\mathbf{c}$ are inequality constraints that enforce desired design conditions.

The single-objective optimization problem is solved with a sequential quadratic programming algorithm, in which gradients are computed by means of finite differences (Mathworks, 2019). The multi-objective optimization problem is solved with a non-dominating sorting genetic algorithm (NSGA-II) (Seshadri, 2020).

This simplified design problem is termed *preliminary*, in the sense that it only determines macroscopic parameters of the machine. Based on the results of this preliminary sizing, standard detailed design procedures should be used to dimension all components and systems (Bortolotti et al., 2016).

## 4   Case study: cost-driven and eco-conscious designs of a wind turbine for Germany

Trade-offs were investigated between an economic and an environmental point of view, by analyzing the characteristics of the resulting optimal turbines with respect to a standard $COE_{\in}$-driven baseline assumed as reference. The study was performed with the methods described in the previous sections, where the cost model was tuned to represent the situation in Germany according to Deutsche WindGuard (2018) and Duffy et al. (2020).

### 4.1 Baseline description

The baseline is chosen to represent a recent COE$_{\text{€}}$-driven industrial product, and corresponds to a wind class IIA machine with a rated power of 3 MW, a rotor diameter of 115.7 m, a hub height of 92 m, and a lifetime of 20 years. These characteristics make the baseline loosely resemble one of the several E-115/3.0 MW models (Enercon, 2021) that, according to Deutsche WindGuard (2018), was the most installed turbine in Germany in 2016, 2017 and 2018 – the years considered in this study. According to the adopted mass models (see §3.2), the rotor blade has a mass of 13 t and a steel tower of 190 t. The main key cost items of the baseline turbine, based on the cost model described in §3.4 and shown in Table 2, are in line with the values provided by Deutsche WindGuard (2018) and Stehly et al. (2017). Slightly lower operating expenditures are reported by Stehly et al. (2017), because of the different location of the study (Duffy et al., 2020).

**Table 2.** Comparison of some key cost items of the baseline turbine with values sourced from the literature.

|  | Baseline | Deutsche WindGuard (2018) | Stehly et al. (2017) |
|---|---|---|---|
| Rated power [MW] | 3 | 2 to 3 | 2.32 |
| Diameter [m] | 115.7 | - | 113 |
| Hub height [m] | 92 | less than 100 | 86 |
|  | Cost [€/kW] | Cost [€/kW] | Cost [€/kW] |
| Rotor | 274 | - | 276 |
| Drivetrain & Nacelle | 400 | - | 469 |
| Tower | 192 | - | 206 |
| Turbine capital expenditures | 866 | 1000 | 951 |
| Balance of station | 343 | 331 | 313 |
| Operating expenditures | 54 | 52 | 38 |

This wind turbine has an COE$_{\text{CO2}}$ of 12.37 kg CO$_2$eq/MWh and an COE$_{\text{€}}$ of 38.6€/MWh, according to the models of §3.4. Given the typical large uncertainties in the discount rate, $d = 0$ was assumed in Eq. (1).

Figure 5 shows a breakdown of the environmental cost of the wind turbine by its principal components. The figure reports both relative emissions with respect to the overall impact produced by the wind turbine (blue bars), as well as absolute emissions per unit of component mass (green bars). Tower and foundations play the largest role in the overall COE$_{\text{CO2}}$, each one accounting for about 20% of the total. The high environmental impact of the foundations is due to the significant amount of concrete that they require, and the negative effects caused by landfilling at the end of life. The tower, on the other hand, is made of steel, a material with a high recyclability rate (see Fig. 2). Notwithstanding the EOL emission credits assumed by the closed-loop material cycle approach, the tower still has a significant environmental impact because of its very large mass. Blades also present a large environmental impact, because of their reduced recyclability. Electronics have the highest impact per unit of material, but a small overall contribution due to their reduced mass.

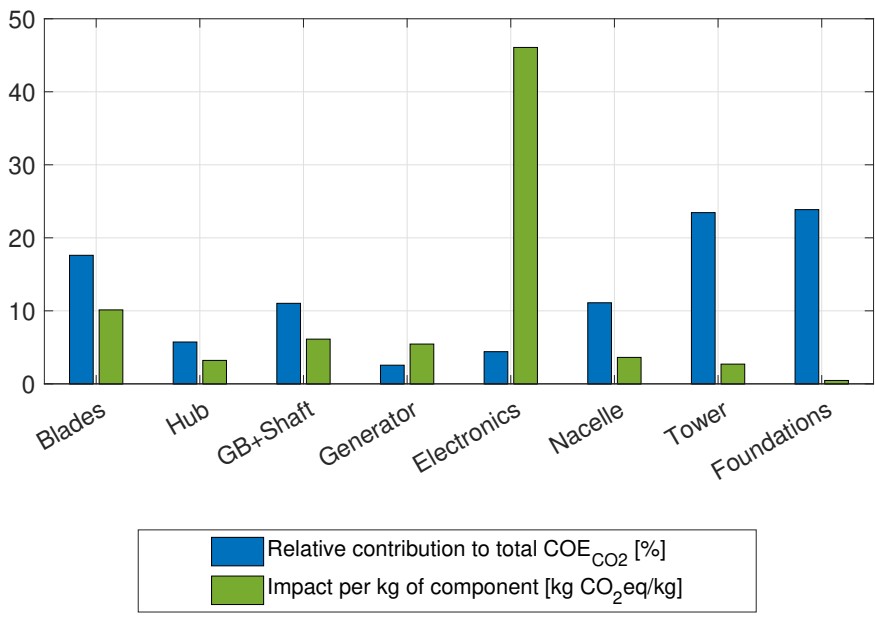

**Figure 5.** Environmental impact of the life-cycle of each component of the baseline wind turbine, expressed in terms of relative percent emissions with respect to the overall impact of the machine (blue bars), and absolute emissions per unit of component mass (green bars).

## 4.2 Cost-driven design

The baseline turbine was then optimized from a combined economic-environmental cost perspective. Only hub height and rotor diameter are free design variables, whereas rated power is held fixed to the baseline value. The bi-objective design problem is expressed by Eq. (15), where $J$ considers economic cost by $COE_{€}$ and environmental cost by $COE_{CO2}$. The design constraints of Eq. (15b) are set to express conditions on the height over diameter ratio and on the specific power of the turbine:

$$0.5 < \frac{H}{D} < 1, \tag{16a}$$

$$100\,\text{Wm}^{-2} < \frac{P_r}{A} < 350\,\text{Wm}^{-2}, \tag{16b}$$

where $P_r$ is the rated power, and $A = \pi D^2/4$ is the rotor swept area. These same inequality constrains were used also in all the following design problems.

Figure 6a shows the resulting Pareto front of optimal non-dominating solutions. The corresponding optimal rotor diameters and hub heights of the Pareto front designs are shown in Fig. 6b.

Results indicate that a decrease in $COE_{CO2}$ can be achieved by reducing the overall size of the turbine, both in terms of rotor diameter and hub height; since rated power is held fixed, the resulting turbines have an increased specific power $P_r/A$. A maximum reduction in $COE_{CO2}$ of about 8% is achieved at the expense of an increase of about 5% in $COE_{€}$.

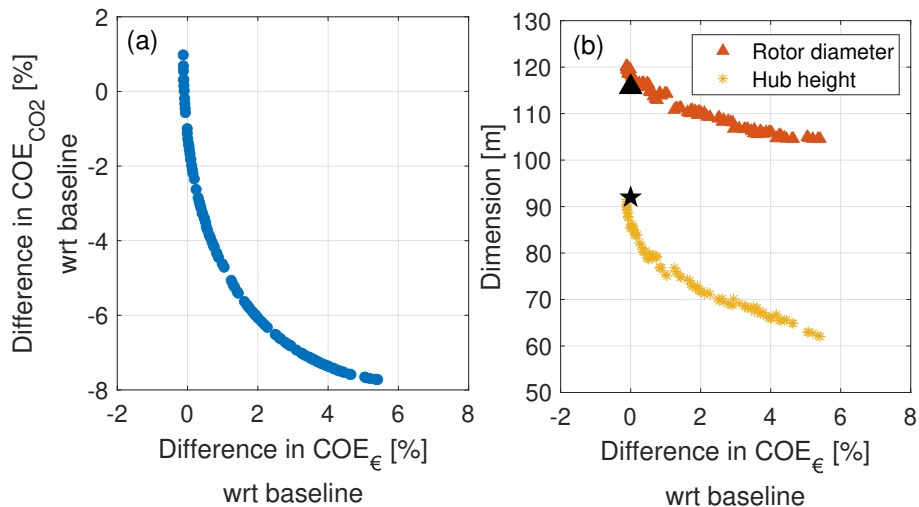

**Figure 6.** Pareto front of $COE_{CO2}$ vs. $COE_{€}$ **(a)**. Rotor diameter and hub height of the Pareto optimal designs **(b)**. Differences are expressed with respect to the baseline configuration, whose dimensions are indicated by black ▲ (diameter) and ★ (hub height) symbols.

However, it is even more interesting to observe that the curve is very steep close to the point of minimum $COE_{€}$. This means that a significant reduction in $COE_{CO2}$ can be achieved with marginal increments in $COE_{€}$. For instance, a turbine with a 110 m diameter and a 75 m hub height presents a $COE_{€}$ that is only 1% higher than the baseline, while at the same time it achieves a $COE_{CO2}$ reduction of about 5%. This result is obtained by the design of smaller rotors and shorter towers that, although imply a somewhat reduced power capture, have lower environmental costs.

### 4.3 Value-driven design

The previous section showed that, from a cost perspective, there is room to reduce the impact on the environment if one is willing to accept some increase in the cost of energy from wind. However, cost by itself does not capture the full complexity of the problem, and further insight can be obtained by including also value in the analysis.

To this end, the turbine was optimized considering economic and environmental value, instead of cost. Two different locations in Germany were selected: one in the north of the country (labelled LN in the following), characterized by very good wind conditions, and a second one in the south (labelled LS), with lower average wind speeds. The site wind characteristics are more precisely shown by the two Weibull distributions reported in Fig. 7 (NEWA, 2021).

The economic and environmental values were estimated with the model described in §3.5. Day-ahead spot market price and energy mix time series were collected from the SMARD database (SMARD, 2020), and completed with wind speed time series obtained from NEWA (2021), considering the years 2016, 2017 and 2018. All quantities were sorted into 50 wind speed bins,

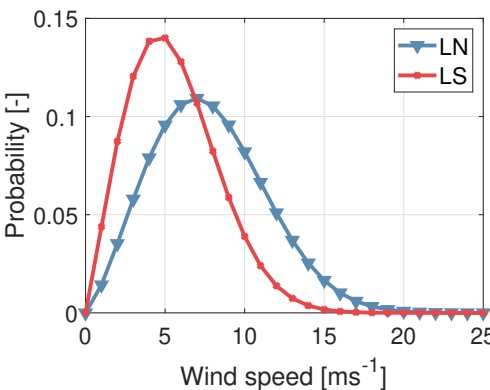

**Figure 7.** Weibull distributions at the northern and southern German locations, at 50 m height above ground.

each containing the same number of data points. A trend curve of cost vs. wind speed was obtained from the mean values of
each bin, and extrapolated above the last bin upper boundary all the way to cut-out wind speed.

Figure 8 and 9 respectively show the curves of spot market price and grid GHG emissions vs. wind speed at 50 m height at
the LN and LS sites for the three considered years. To reduce clutter, the figures only show 9 of the 50 bins used to create the
450 curve. For both locations, the spot market price and grid GHG emissions exhibit a decreasing trend with respect to wind speed.
In fact, at low wind speeds there is a large amount of energy from coal-fired power stations in the energy mix, pushing both
the price and grid GHG emissions up. With higher wind speeds, the amount of wind energy in the grid increases, so that more
expensive and polluting energy sources are displaced. This is clearly a partial view of the behavior of a very complex system,
which does not only depend on wind speed.

### 4.3.1 Single-objective optimization

First, a single-objective optimization was run for each metric at the two locations in order to analyze the behavior of the optimal
turbine design characteristics. The resulting diameters are shown in Fig. 10a, while the hub heights are given in Fig. 10b. The
figures of merit are organized from left to right as follows: the first two are cost-based metrics ($COE_{€}$ and $COE_{CO2}$), the next
two are value-based metrics ($VOE_{€}$ and $VOE_{CO2}$), and finally the last two are net-value-based metrics that consider both cost
and value ($NVOE_{€}$ and $NVOE_{CO2}$).

Analyzing first the cost-based perspective, results indicate that, as already observed in §4.2, a turbine designed for minimum
$COE_{CO2}$ has a smaller rotor and a shorter tower than a turbine designed for minimum $COE_{€}$, on account of their large environmental impact. For both metrics, the southern location LS requires a turbine with a larger rotor and a taller tower than the
northern location, due to lower typical wind speeds.

From a value point of view, no differences in rotor diameter and hub height are found between the economic ($VOE_{€}$) and
the environmental ($VOE_{CO2}$) perspectives. In fact, for both metrics, the optimal rotor and hub height are as large as possible,
hitting the lower bound for specific power. This can be explained by noticing that, since low wind speeds are associated with

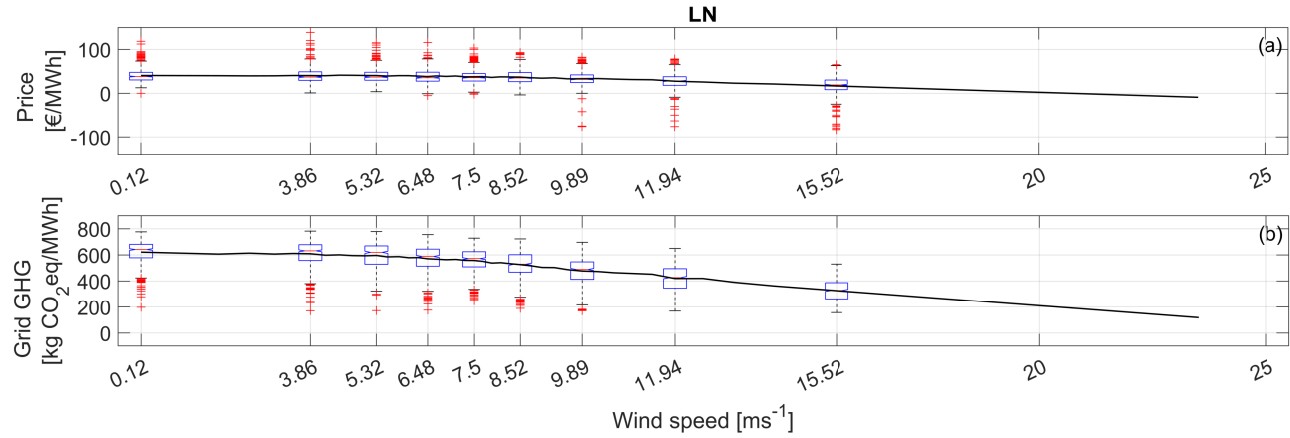

**Figure 8.** Spot market price **(a)** and grid GHG emissions **(b)** vs. wind speed at 50 m for LN (site in the north of Germany). In the boxplots, the central red mark indicates the median, and the bottom and top blue edges of the box indicate the 25th and 75th percentiles, respectively. Whiskers extend to the most extreme data points not considered as outliers, whereas outliers are plotted individually using a red '+' symbol. Above 15.52 ms$^{-1}$, the curve is extrapolated.

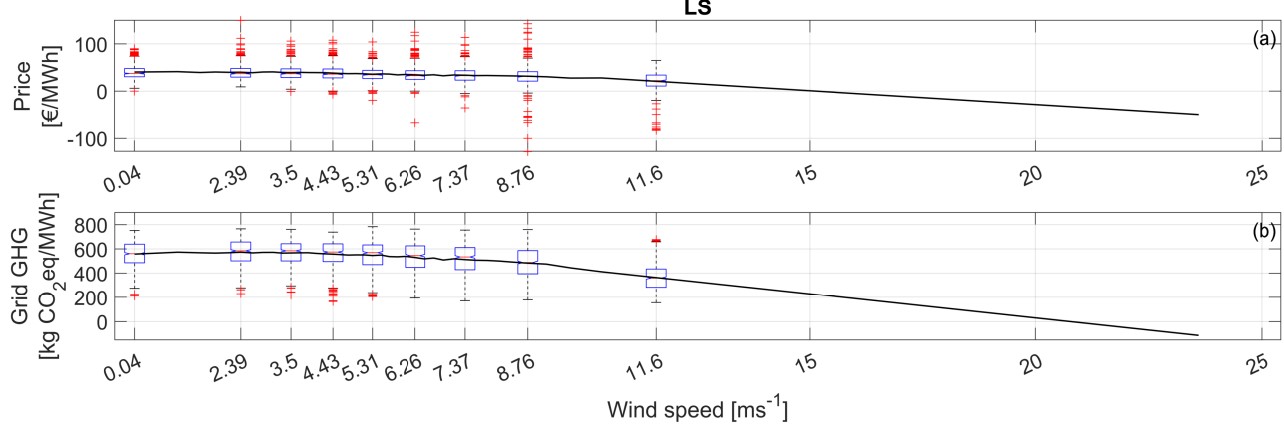

**Figure 9.** Spot market price **(a)** and grid GHG emissions **(b)** vs. wind speed at 50 m for LS (site in the south of Germany). The data is plotted as in Fig. 8, with the extrapolation starting at 11.6 ms$^{-1}$.

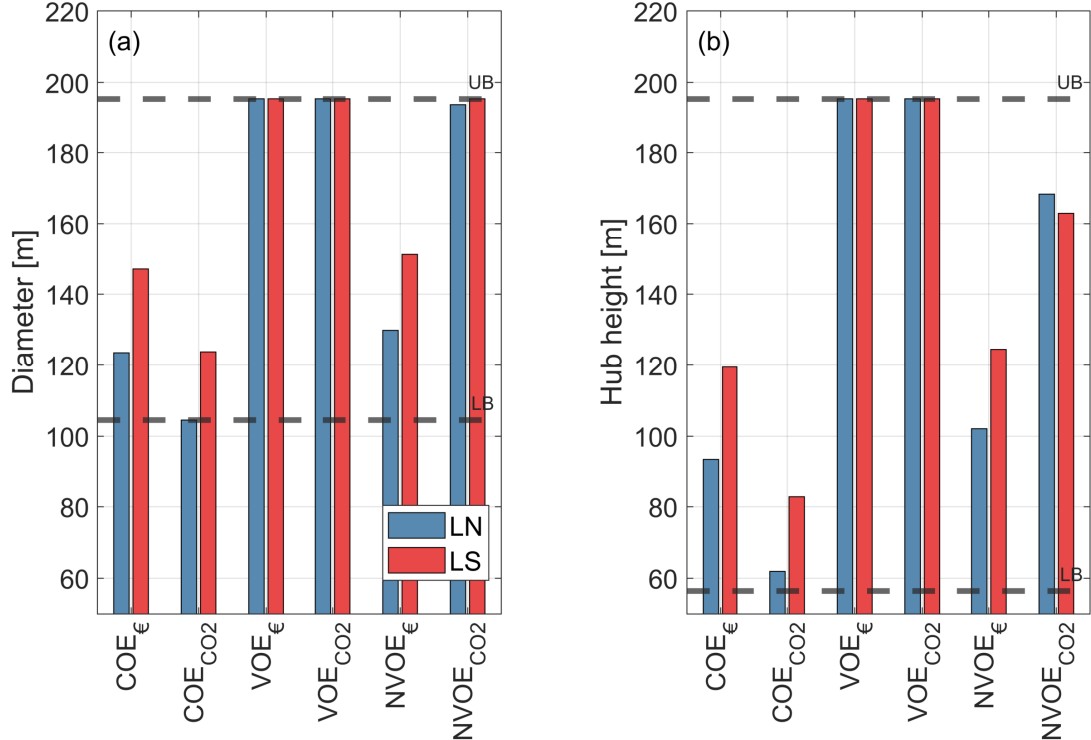

**Figure 10.** Optimal diameters **(a)** and hub heights **(b)** for each single objective function, for the two locations. Cost-based metrics: $COE_{€}$, $COE_{CO2}$; value-based metrics: $VOE_{€}$, $VOE_{CO2}$; net-value-based metrics: $NVOE_{€}$, $NVOE_{CO2}$.

larger economic and environmental values (see Figs. 8 and 9), optimal economic/environmental value-driven designs tend to produce as much as possible at low wind speeds. This can be achieved by minimizing the extent of the partial load region
(region II), which is obtained by reducing the rated wind speed $V_r = \sqrt[3]{2P_r/\rho A C_{P_{\max}}}$, where $\rho$ is the air density and $C_{P_{\max}}$ the maximum power coefficient of the rotor. As shown by the formula, since $C_{P_{\max}}$ is limited by physics, for given ambient conditions $\rho$, $V_r$ decreases for smaller specific powers $P_r/A$. These results are in line with similar studies that have shown how low specific-power turbines have a higher economic value (Hirth and Mueller, 2016; Swisher et al., 2022).

Finally, both $NVOE_{€}$ and $NVOE_{CO2}$ – which consider both cost and value – lead to configurations that can be interpreted
as compromises between the cost and value perspective. For $NVOE_{€}$, as economic value has the same order of magnitude as economic cost, the solution presents a rotor diameter and hub height that fall in between the cost- and value-based solutions. On the other hand, for $NVOE_{CO2}$ the environmental value is one order of magnitude larger than the environmental cost, and this drives the rotor size, which even in this case hits the lower limit for specific power. The introduction of cost, however, penalizes the hub height, which is reduced with respect to the value-based solution because of the large influence of the tower.

### 4.3.2 Bi-objective optimization

Next, trade-offs between the economic and environmental net value were analyzed by examining the Pareto front obtained by solving the bi-objective optimization problem expressed by Eq. (15). The $COE_{\text{€}}$-driven designs of the previous section and displayed in Fig. 10 are used here as baselines for each location.

For the two sites LN and LS, Fig. 11a shows the Pareto front $NVOE_{CO2}$ vs. $NVOE_{\text{€}}$, while Fig. 11b reports the change in rotor diameter and hub height with respect to the baselines, as functions of $NVOE_{\text{€}}$. As already observed in Fig. 6, even in this case results indicate that it is possible to increase the environmental net value ($NVOE_{CO2}$) without significantly decreasing the economic net value ($NVOE_{\text{€}}$). For example, accepting a decrease in $NVOE_{\text{€}}$ of 1 €/MWh buys half of all possible improvement in $NVOE_{CO2}$, for both locations. This is achieved with larger diameters (i.e., smaller specific powers), and taller hub heights.

Another interesting observation is that both locations present the same Pareto front shape. While LN has a better economical performance than LS (as expected, because of the better wind resource), both locations appear to have a similar net value from an environmental point of view.

Finally, environmental net value was used to estimate future economic societal savings, multiplying $NVOE_{CO2}$ by SCC, as described in §2.2.4. An SCC of 1 €/kg $CO_2$eq was considered in this work. However, as previously noted, SCC can take widely different values depending on the assumptions and models considered (IPCC, 2007). Although this makes the resulting FSS values affected by high uncertainty, the analysis is still useful because it may reveal interesting trends.

Figure 12 presents the designs that result from trading $COE_{\text{€}}$ – the metric currently used to asses the competitiveness of an energy-producing technology – with FSS – the metric proposed here to estimate the future societal savings obtained by deploying an energy-producing technology.

The Pareto front is displayed in absolute quantities in Fig. 12a, and relative to the $COE_{\text{€}}$-driven baselines in Fig. 12b. Similarly, the solutions of the Pareto front are displayed in absolute quantities in Fig. 12c, and relative to the baseline configurations in Fig. 12d. The values shown here should be treated only as rough estimates because of the many simplifications and assumptions. Nonetheless, some interesting trends seem to emerge.

First, as expected, the current $COE_{\text{€}}$-driven designs (which capture the individual point of the view of the investor and consumer) are not optimal from the societal point of view. This means that, to improve the societal metric, an individual would have to accept an increase in out-of-pocket expenditure.

Second, the largest opportunities appear to be close to the $COE_{\text{€}}$ optima, where the curves are very steep. This means that even marginal increases in cost can have an impact on societal savings. However, away from the $COE_{\text{€}}$ optima, the curves level off, meaning that optimal societal savings would require significant increases in cost, which would probably not be acceptable by consumers.

Third, the general trend of the Pareto solutions is similar at both sites. Hence, even at sites characterized by poor wind resources, there is room for improving the societal value of wind energy.

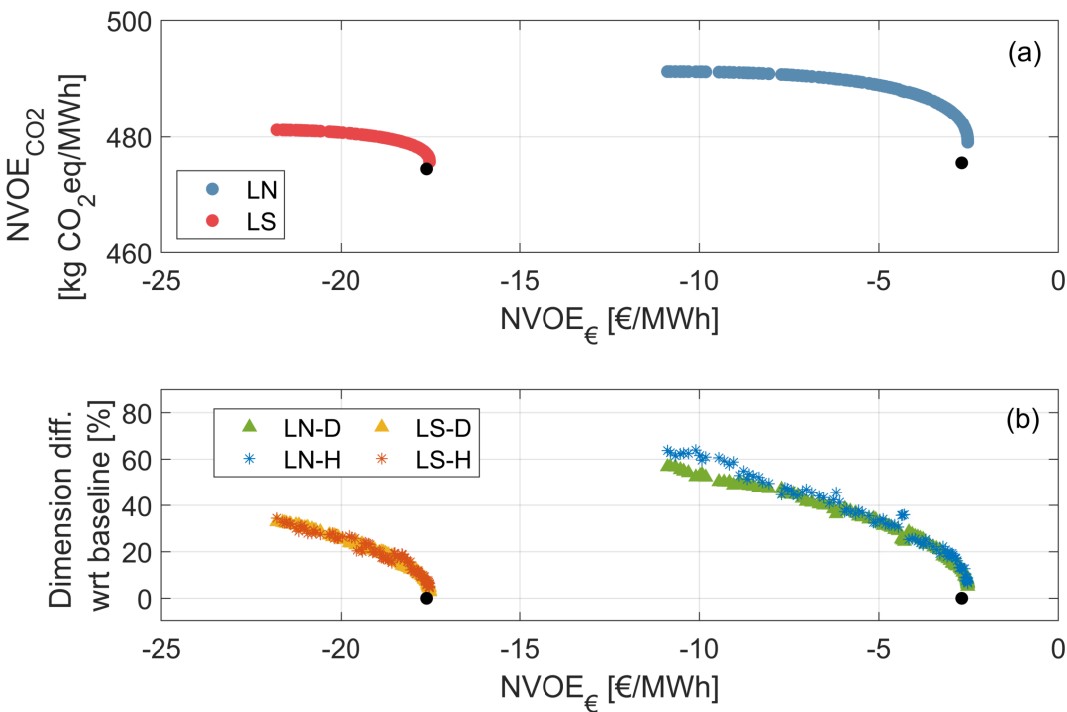

**Figure 11.** Pareto front between a net environmental value point of view (NVOE$_{CO2}$) and a net economic value point of view (NVOE$_€$) **(a)**. Optimal diameters and hub heights for the solutions of the Pareto front, expressed as percent changes with respect to the corresponding COE$_€$-driven baseline of each location **(b)**.

Fourth, although better wind resources at the site in northern Germany are associated with lower costs, the societal savings are similar at both locations. This is an interesting finding, because it implies that the installation of each new wind turbine is of a similar environmental and societal value, independently of the characteristics of the site. However, since sites with worse wind resources are penalized by a higher COE$_€$, policies may be needed that – by taking a long-term view on future economic societal savings – increase in the short-term the competitiveness of wind turbines at these locations.

## 5 Conclusions

This paper has explored the idea of enhancing the inherent environmental and societal value of wind turbines by changing the way they are designed. While "value" is clearly a very broad concept, and multiple metrics would be needed to capture its many facets, the focus here is on the benefits brought by the displacement of climate-changing environmental emissions. To some extent, these environmental effects can be turned into societal ones by using the economic cost of the future damage that they will cause.

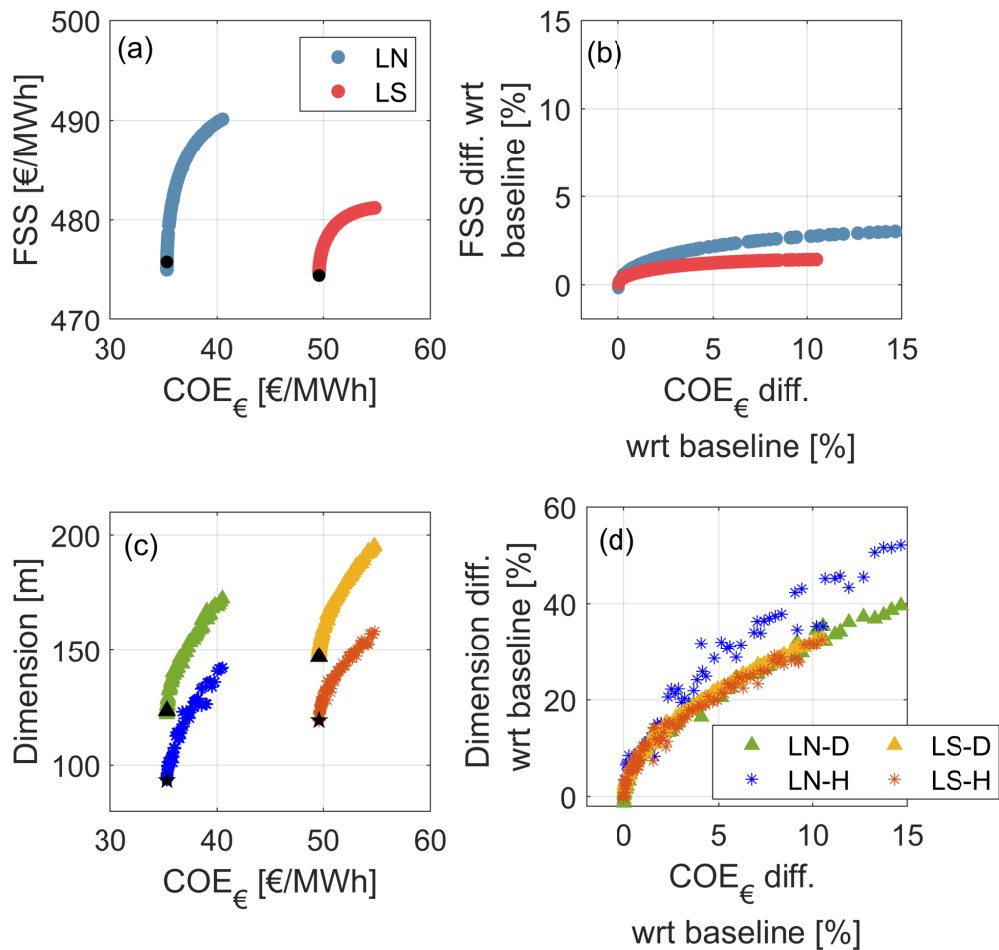

**Figure 12.** Pareto front between FSS and COE$_€$, in absolute values (**a**), and relative to the COE$_€$-driven baseline of each location (**b**). Optimal diameters and hub heights of the Pareto solutions in absolute values (**c**), and relative to the corresponding COE$_€$-driven baseline of each location(**d**).

The paper first defined two new concepts: *environmental cost* and *environmental value*. The former expresses the GHG emissions generated throughout the entire lifetime of a wind turbine, while the latter quantifies the displacement of GHG emissions from the grid caused by the deployment of a wind turbine; in both cases, these quantities are computed per unit of generated energy. These metrics are defined mirroring already existing economic metrics, based on the familiar concepts of economic cost and value.

Next, a toolchain was described, which implements a complete LCA model capable of estimating the emissions of a wind turbine throughout its lifetime, broken down in life-cycle stages, components and materials. Using the LCA model, together with energy and mass models, a simplified design problem was formulated, which can determine the optimal geometric characteristics of a wind turbine (in terms of its rotor diameter and hub height) for a given rated power. The resulting preliminary design gives only the overall dimensions of the turbine, and would have to be followed by a detailed design of its aerodynamics, structures, systems and control laws. The design problem can be formulated either as a single or a multi(bi)-objective minimization. The classical standard approach of designing wind turbines by minimizing $COE_{\text{€}}$ is included in the formulation as a special case.

A 3 MW wind turbine, representative of the $COE_{\text{€}}$-driven machines currently installed in Germany, was chosen as a baseline reference to study the effects of considering various possible economic and/or environmental metrics in the preliminary sizing problem.

The environmental assessment of the baseline highlighted the large contribution of the foundations – made of concrete –, and of the tower – made of highly-recyclable steel – to the total life-cycle emissions of the wind turbine. These components have low emission factors – i.e., a low environmental cost per unit of mass – but require large quantities of material. Electronic components, on the other hand, have a modest overall contribution even if they have very high emission factors. Clearly, the overall environmental cost of a wind turbine depends on the technological solutions chosen for its main components. Indeed, choices at all stages of the life-cycle – from the processes used to mine or produce the materials to EOL decisions – have implications on both the economic and environmental costs, as the two aspects are intimately connected. Understanding the environmental cost of each material, each component and each life-cycle stage is critical for identifying alternatives that minimize both forms of cost.

The baseline turbine was then redesigned using a bi-objective optimization for $COE_{\text{€}}$ and $COE_{\text{CO2}}$, obtaining a Pareto front of optimal non-dominating solutions. This family of solutions can be interpreted as the cost-optimal designs that trade the point of view of the individual ($COE_{\text{€}}$) with the point of view of the environment ($COE_{\text{CO2}}$). It is one of the main findings of this work that the Pareto front is very steep around the $COE_{\text{€}}$-optimal designs. For the case considered here, it appears that a $COE_{\text{€}}$ increase of only 1% can buy a $COE_{\text{CO2}}$ decreases of 5%. In other words, it pays off to be altruistic, and a large reduction of the environmental impact can be achieved if consumers are willing to pay a bit more for the energy that they consume.

Finally, the effects of value and net value were considered, again looking at both the economic and environmental points of view. Value-based metrics are location- and time-dependent quantities, and therefore tightly linked to the site where the wind turbine is installed. Two locations were considered: one in the north of Germany with better wind resources, and one in the

south of the country, where typical wind speeds are lower. Results show that, for the years considered here, spot market price and grid GHG emissions are generally higher at low wind speeds for both sites, as generally expected.

A Pareto front of optimal solutions was generated that trades-off economic net value – i.e. the difference between economic value and cost –, and environmental net value – similarly defined, but considering emissions. Results indicate that, here again, the curves are very steep close to the net-value economic optima. Therefore, even from this point of view altruism pays off, and significant net value environmental gains can be achieved with rather small losses in net economic value.

Unsurprisingly, economic net values were found to be profoundly different at the two locations, the better wind resources
in the north being associated with much lower values of $NVOE_€$. However, interestingly, the environmental net values at the two locations were found to be very similar. This result points to the fact that wind turbines have similar beneficial effects no matter where they are installed, with little sensitivity to the local wind resources (at least for the present German scenario). Therefore, wind energy is a sensible choice also for places with modest wind conditions, as for example the south of Germany. These results should be further explored considering transmission constraints.

Additionally, it was found that environmental value is one order of magnitude larger than environmental cost, whereas economic value and cost are of the same order of magnitude. Consequently, the economic net value is more sensitive than the environmental one to the characteristics of the location. This conclusion, here again, is valid only for the specific electricity market of Germany. Further studies should analyze the environmental and economic value of wind turbines in different electricity markets, for example with a smaller or larger penetration of renewables.

Finally, future societal savings were estimated by using the societal cost of carbon, which quantifies the present cost of future damage caused by the emission of one additional unit of $CO_2$eq. Similar conclusions as the ones discussed earlier can be drawn from these results.

This study shows that, in general, low specific-power turbines present higher economic and environmental values, at the expense of a higher cost of energy. This is due to the fact that, with the present technology, the larger energy captured by
a bigger rotor does not generally compensate its larger cost. However, the present findings highlight that the benefits of low specific-power turbines go well beyond what is quantified through $COE_€$ alone, which, in hindsight, appears to be a rather myopic and incomplete metric. Indeed, several studies have shown that low specific-power turbines bring benefits beyond economic value: for instance they can better utilize the transmission system, reduce forecasting errors, and could lead to cheaper financing (Hirth and Mueller, 2016; Swisher et al., 2022).

The present work and its findings are affected by several limitations.

First, the LCA, mass, and cost models are based on general trends of current wind turbines. Clearly, low specific-power machines push the boundaries of these models. More accurate estimates could be obtained by using detailed design procedures (for example, see Bortolotti et al. (2016); NREL (2021)) that, from the rough sizing produced by the present approach, yield refined designs. Additionally, the study could be extended to investigate the impact of additional variables; one such example
is rated power, which was assumed here to be given and fixed, but in reality could be freed to possibly reveal other features of the solution space.

The trends shown here are only valid for Germany in the years considered. Clearly, both economic and environmental value depend on the time-specific composition of the energy mix, whose behavior is very complex and depends on more variables than just wind speed, as it was assumed here for simplicity. The assumptions taken in this work are clearly oversimplifications that try to produce initial rough preliminary trends. Future work should couple the present models with more sophisticated descriptions of the energy mix, able to capture their present and future composition. In fact, understanding how the economic and environmental value of wind energy will develop in the future is yet another crucial element that deserves further work. Indeed, as wind penetration is set to increase, the economic value of wind energy is expected to decrease, an effect called "self-cannibalization". However, predicting the impact of an increase in wind energy is not straightforward, as the final effects depend on the emission factors of the generating technologies in the energy mix. The impact on displaced GHG is even more complex to estimate, as it depends on the emission factors of the generating technologies operating on the margin, which are not only strongly country-specific, but also time-dependent. Here again, these effects can only be properly captured by using more sophisticated models, including an electricity market model.

The results presented in this work are subject to significant uncertainties. Indeed, in addition to the uncertainties brought by the variable nature of the wind resource, one should also consider the uncertainties brought by the volatility of the electricity market, and the uncertainties in the LCA model (which are significant, given the holistic nature of a life-cycle analysis, with its many stages and potentially complex processes). While mean yearly values were used in this preliminary work for simplicity, further studies should be conducted from a probabilistic point of view – for instance through uncertainty quantification methods – to produce a more detailed picture of the variability of the economic and environmental cost and value of wind turbines.

Notwithstanding these limitations, it was one major ambition of this paper to bring the inherent environmental and societal value of wind turbines under the spotlight. Indeed, the paper shows that a purely economic analysis paints only a very partial picture of the true nature and possible role of a wind turbine (and, more in general, of wind energy). Indeed, enlarging the perspective away from economics can uncover new opportunities for the future development of wind. While this study only focused on the changes in overall dimensions (and, in turn, specific power) of the machine, the potential for further improvements is much larger that what would appear by this simple analysis alone. In fact, the same metrics developed here can also be employed to guide the choice of technologies and the detailed design of the various components of a wind turbine. In addition, beyond the single wind turbine case analyzed here, this new eco-conscious design philosophy can be used to design a whole wind plant.

*Author contributions.* HC led the development of the work, in close collaboration with AG and CLB. AG developed and validated the LCA model. CLB supervised the research. HC and CLB wrote the paper, with inputs from AG. All authors provided important input to this research work through discussions, feedback and by writing the paper.

*Competing interests.* The authors declare that they do not have conflicts of interest, except for CLB who is the Editor in Chief of WES.

*Acknowledgements.* The authors acknowledge the participation of Samuel Kainz and Guillermo Fuente Taravillo, both from the Technical
University of Munich, the former for the revision of the LCA model, and the latter for the collection of wind speed data for different locations
in Germany and for input in the early stages of the work.

**Nomenclature**

|  |  |  |
|---|---|---|
| | $A$ | Rotor swept area |
| | $C$ | Cost |
| 630 | $D$ | Rotor diameter |
| | $E$ | Energy |
| | $H$ | Hub height |
| | $J$ | Cost function |
| | $P$ | Power |
| 635 | $T$ | Duration |
| | $Q$ | Emissions |
| | $U$ | Wind speed |
| | $V$ | Value |
| | $W$ | Weibull distribution |
| 640 | $\mathbf{c}$ | Constraints |
| | $d$ | Discount rate |
| | $f$ | Factor |
| | $p$ | Spot market price |
| | $\mathbf{p}$ | Design parameters |
| 645 | $t$ | Time |
| | $(\cdot)_y$ | Relative to year $y$ |
| | $CO_2eq$ | Equivalent mass of $CO_2$ with the same global warming potential of a given gas |
| | $COE_{\text{€}}$ | Economic levelized cost of energy |
| | $COE_{CO2}$ | Environmental cost of energy (in terms of climate-changing $CO_2eq$ emissions) |
| 650 | EOL | End of life |
| | FSS | Future societal savings |
| | GFRP | Glass-fiber reinforced plastic |
| | GHG | Greenhouse gases, i.e. $CO_2$, $CH_4$, $NO_2$, F-gases, among others |

| | LCA | Life-cycle assessment |
|---|---|---|
| 655 | NVOE$_\epsilon$ | Economic levelized net value of energy |
| | NVOE$_{CO2}$ | Environmental net value of energy (in terms of climate-changing $CO_2$eq emissions) |
| | SCC | Societal cost of carbon |
| | VOE$_\epsilon$ | Economic levelized value of energy |
| | VOE$_{CO2}$ | Environmental value of energy (in terms of climate-changing $CO_2$eq emissions) |

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
