# Peer review of "The eco-conscious wind turbine: design beyond purely economic metrics"

_Wind Energy Science, 2022_

## Author Comment (AC1)

**REVISION TO MANUSCRIPT DRAFT**

**Wind Energy Science Discussion**

**The eco-conscious wind turbine: bringing societal value to design**

The authors would like to thank the two reviewers for their time and for the useful feedback. All inputs that they provided have contributed to the improvement of the paper.

A list of point-by-point replies to the reviewers' comments is reported in the following.

We have taken the opportunity to make several small editorial changes to the text, in order to improve readability. A revised version of the manuscript is attached to the present reply, with additions highlighted in blue and deletions marked in red.

The authors
* * *
**Reviewer #1**

*[Reviewer] This is a great paper that certainly deserves publication in WES. I'd like to congratulate the authors for their hard work and I only have a couple minor comments to further improve their article.*

**Numbered comments**

1. *[Reviewer] Section 2.1: there are several more LCOE+ metrics in literature than the ones that you report here. I miss why you chose LVOE and NVOE opposed to others, for example PLCOE, which is recommended by Mai et al, 2021.*
   **[Authors]** The goal of this paper is to emphasize the mirroring between an economic and environmental perspective and define eco-conscious metrics to quantify the environmental impact and benefits brought by a wind turbine, which can be applied to guide an optimization exercise. Even though we are aware that there are multiple LCOE+ metrics in the literature, we chose to focus on LVOE and NVOE because these metrics are defined similarly to the classic and very well-known LCOE figure, and rely only on the concepts of cost and value, which are also the main focus of this paper. Clearly further research should focus on further extending the list of eco-conscious metrics. We have added a paragraph at the beginning of Section 2 to emphasize the existence of other metrics, citing three relevant references from the recent literature.

2. *[Reviewer] Sections 3.2 and 3.4 should be expanded. I understand that you are scaling masses and costs solely from rotor diameter and hub height. Your inputs must also include fixed quantities such as rated power and max tip speed (?), which help estimate gearbox and generator torque. A couple extra sentences would help. Also, to show the validity of the assumptions, you should report masses and costs for the baseline WT and show that the absolute values match reasonably well with literature, for example with turbine capital cost numbers provided in https://www.nrel.gov/docs/fy22osti/81209.pdf*

   **[Authors]** The mass model is based on the NREL Cost and Scaling Model 2017, which estimates the mass of the different wind turbine components based on hub height, rated power and rotor diameter, as illustrated in Fig. 1. The gearbox mass is estimated based on the maximum generator torque, which is computed considering a fixed maximum tip speed of 80 m/s. The cost model

estimates the different cost items based on the masses of the different components. Figure 1 shows the dependency of the cost of the different components with respect to the inputs.

[Figure]

*Figure 1.* *Visualization of the dependency of masses and costs with different inputs of the mass and cost model.*

The mass model validation has been performed in comparison to several documents from the recent literature. The blade mass of the baseline considered for this study and described in Sect. 4.1 has a rotor diameter of 115.7 m and a mass of 12.72 tons, which is comparable to the 16 tons of the IEA Task 37 130m land-based rotor[1]. The tower of this baseline turbine presents a mass of 190 tons, which matches well the mass of towers under 100 m described in the report *Increasing Wind Turbine Tower Heights: Opportunities and Challenges*[2].The cost model is validated against the values indicated in Stehly et. al[3], WindGuard[4] and Duffy et al.[5]. The costs of the baseline show a good agreement with the costs of similar projects in Germany, as indicated by WindGuard. The slightly higher turbine capital (TCC) expenditures result from differences in the assumptions, with WindGuard considering also transportation and installation costs within TCC. Stehly et al. also shows similar costs with respect to the baseline. Slightly higher operating expenses are found for the baseline and WindGuard values, since Germany presents higher average operating expenses than the US for the years of the study, as shown in Duffy et al.

**Table 1.** Overview of the mass and costs of the baseline, and similar reference turbines used in WindGuard & Stehly et al.

| Baseline | | | WindGuard | Stehly et al. |
|---|---|---|---|---|
| Rated power [MW] | 3 | | 2 to 3 | 2.32 |
| Diameter [m] | 115.7 | | | 113 |
| Hub height [m] | 92 | | Less than 100 | 86 |
| | **Mass [t]** | **Cost [€/kW]** | | **Cost [€/kW]**[6] |

[1] Bortolotti et al. NREL/TP-5000-73492

[2] Lantz E. et al.: Increasing Wind Turbine Tower Heights: Opportunities and Challenges. NREL Report/TP-5000-73629, May 2019

[3] Stehly T. et al.: 2017 Cost of Wind Energy Review. NREL Report/TP-5000-73629, May 2019

[4] DeutscheWindGuard: Wissenschaftlicher Bericht. Vorbereitung und Begleitung bei der Erstellung eines Erfahrungsbericht gemäßt §97 Erneuerbare-Energien-Gesetz. Teilvorhaben II e): Wind an Land

[5] Duffy et al.: Land-based wind energy cost trends in Germany, Denmark, Ireland, Norway, Sweden and the United States, https://doi.org/10.1016/j.apenergy.2020.114777

[6] Considering a dollar/euro conversion ratio of 1.15.

| | | | | |
|---|---|---|---|---|
| **Rotor** | 7.7361e+04 | 274 | | 276 |
| **DT+Nacelle** | 1.1929e+05 | 400 | | 469 |
| **Tower** | 1.9065e+05 | 192.33 | | 206 |
| **Turbine Capital Expenditures** | | 866.33 | 1000 | 951 |
| **BOS** | | 343 | 331 | 313 |
| **Operating expenditures** | | 53.6 | 52 | 38 |

We have now added a paragraph in Sect. 4.1., describing the key cost items of the baseline wind turbine and how they compare with values sourced from the literature.

3. *[Reviewer] Page 12, line 322: "A representative scenario of 50% incineration and 50% landfilling is assumed here, as described in Vestas (2011, 2013a, b)." This is surprising to me, I thought that the vast majority of blades ended up in landfills. I looked at some references, for example https://doi.org/10.1016/j.rser.2021.111847 and https://doi.org/10.1177/1048291116676098, and I struggle to find hard numbers. Probably, percentages change from country to country. This said, the references that you provide also don't seem very solid. Some extra literature and possibly a couple more sentences are recommended to support your assumption.*
   **[Authors]** End of life treatment is very dependent on the specific location of the turbine and the legislation of the country. Different authors take different assumptions for the treatment of the blades, for instance Haapala (US, 2014) and Taskin (Turkey, 2013) consider that they are landfilled, Bang et al. (US, 2019) assume that glass fibers are 100% incinerated, and Meunier (France, 2009) assumes that 98% of the blades are recycled.
   According to WindEurope[8], landfill is essentially not used anymore in some European countries (i.e. Germany, Austria, Belgium and Denmark), and composite materials waste is banned altogether from landfills in Germany and in the Netherlands[7]. On the other hand, there are currently no landfill bans in any of the US states for composite waste and wind turbine blades. We updated the scenario considered in this study to better adhere to the German legislation, which bans the landfilling of composites. The actual end of life of blades in Germany is complex, as options beyond landfilling, recycling or incineration are possible; for instance, wind turbine blades are often used in cement co-processing[8]. However, for simplicity and due to the lack of representative data, a scenario of 100% incineration is now considered in the revised paper.

4. **[Reviewer]** *Figure 9: why is the y axis so tiny? I cannot interpret this plot: I do not see the drop in price with wind speed and I don't understand what the red markers represent (is it a box-whisker plot?). The caption doesn't help me much either.*
   **[Authors]** The plot shows the data for price and grid GHG distributed in bins. The format chosen is a box plot, where the central mark indicates the median, and the bottom and top edges of the box indicate the 25th and 75th percentiles, respectively. The whiskers extend to the most extreme data points not considered as outliers, whereas outliers are plotted individually using the red '+' symbol. We have updated the caption to more precisely describe the boxplots. Additionally, we
* * *
[7] Beauson et al. *https://doi.org/10.1016/j.rser.2021.111847*
[8] WindEurope *https://windeurope.org/wp-content/uploads/files/about-wind/reports/WindEurope-Accelerating-wind-turbine-blade-circularity.pdf*

have updated the plot and added a line that follows the mean value of each boxplot and is extrapolated all the way to cut-out speed.

**Reviewer #2**

[Reviewer] *The manuscript of Canet et al. on "The eco-conscious wind turbine: bringing societal value to design" is a timely and important contribution that shows a way forward how to quantify and trade the value of wind energy beyond the economical cost of energy. This way it can facilitate discussions beyond speculation and preconceived notions.*

**Numbered comments**

1.  [Reviewer] *There a key insights in the work that may be stressed even more clearly by the authors in the abstract and conclusions than they already do:*
    *- similar as in economic metrics, like LCoE, one needs to look at the difference between value and costs (and not costs alone), here it is that wind energy also displaces $CO_2$ production by an order of magnitude more that it produces.*
    *- "value-based metrics are location- and time-dependent quantities", so here the merit order in the electricity market needs to be accounted for to quantify the $CO_2$ displacement effect.*
    *- There are likely trades possible at little economic costs, or even none, that benefit society at large if quantified and traded in design, e.g. via multi-disciplinary design analysis and optimization (MDAO).*
    [Authors] Thank you for your comments, we have added a final paragraph in the conclusion to emphasize the key messages of the paper.

2.  [Reviewer] *The authors are very much aware of the limitations of their study, but here a few points to consider, although these likely make their conclusions rather stronger:*
    *- In their MDAO, rating of the turbines was kept constant. This is reasonable at first, but when a larger rotor was found to be beneficial for societal impact, some economic penalty (compared to a pure LCoE optimiztion) had to be paid. However, for this larger rotor, a larger rating may then pay off for LCoE.*
    [Author] We agree that it might be interesting to analyze the impact of changing the rating. We have updated the conclusions to clearly state the need for further studies in this direction.

3.  [Reviewer] *- Often their optimization led to an optimum design at boundaries, e.g. the lowest specific power allowed. Besides the question what bounds to choose, here, allowing for a variable rating could also help.*
    [Author] The bounds were defined to stay within the validity range of the cost model. Clearly, further studies beyond these boundaries with more realistic mass and cost models should be performed. We updated the conclusions section to highlight this important point.

    [Reviewer] *- Sensitivity analysis and uncertainty quantification will be important for (future) robust designs using their methodology.*
    [Authors] We agree with the need to further develop this work from a probabilistic point of view. We have updated the conclusion section to emphasize this point.

    [Reviewer] *- The authors found the "environmental net value at the two locations (...) very similar". Is this due to being in the same electricity market with the same merit order?*

**[Authors]** We believe this is the case. Future studies should analyze the effects brought by different electricity markets. We updated the conclusion section to highlight this point.

*[Reviewer] Overall, I consider this well-written and well-structured manuscript a significant contribution to the literature and are looking forward to follow up work by the authors, our research community and beyond!*
Thank you for your comment.

[revised manuscript text omitted]

---

## Author Response (AR2)

**REVISION TO MANUSCRIPT DRAFT**

**Wind Energy Science Discussion**

**The eco-conscious wind turbine: bringing societal value to design**

Dear Associate Editor,

In this revised version we have fixed a few small issues:

- The label of fig. 5 was changed to "at 50 m", while it was previously erroneously indicated as "at 50 ms^-1".
- In fig. 12, the range of LCOE change was excessive.
- Additionally, the colors used for figs. 6, 11 and 12 were inconsistent and have now been modified.

Additionally, we noticed that we had often used the adjective "societal" when in reality referring to effects caused on the environment. To improve clarity, we have rephrased several sentences and also changed the title. Additionally, we have changed the name of a metric from "Impact of Energy" to "Environmental Cost of Energy", because this helps highlight the parallelism between economic and environmental metrics. We updated the notation and figures accordingly, and -for extra clarity- we have also included a new table (no. 1 in the new manuscript) reporting all metrics considered in the paper.

All changes are highlighted in the attached document, with additions marked in blue and deletions in red.

Thank you, and best regards

The authors (Helena Canet, Adrien Guillore, Carlo Bottasso)